# Spin Precession in the Gravity Wave Analogue Black Hole Spacetime

Chandrachur Chakraborty [1,2,*] and Banibrata Mukhopadhyay [1]

[1]  Department of Physics, Indian Institute of Science, Bengaluru 560012, India; bm@iisc.ac.in
[2]  Manipal Centre for Natural Sciences, Manipal Academy of Higher Education, Udupi 576104, India
[*]  Correspondence: chandrachur.c@manipal.edu

**Abstract:** It was predicted that the spin precession frequency of a stationary gyroscope shows various anomalies in the strong gravity regime if its orbit shrinks, and eventually, its precession frequency becomes arbitrarily high very close to the horizon of a rotating black hole. Considering the gravity waves of a flowing fluid with a vortex in a shallow basin, which acts as a rotating analogue black hole, one can observe the predicted strong gravity effect on the spin precession in the laboratory. Attaching a thread with the buoyant particles and anchoring it to the bottom of the fluid container with a short-length miniature chain, one can construct a simple *local* test gyroscope to measure the spin precession frequency in the vicinity of the gravity wave analogue black hole. The thread acts as the axis of the gyroscope. By regulating the orbital frequency of the test gyroscope, one can also measure the strong gravity Lense–Thirring effect and geodetic/de-Sitter effect with this experimental set-up as the special cases. For example, to measure the Lense–Thirring effect, the length of the miniature chain can be set to zero, so that the gyroscope becomes static. One can also measure the geodetic precession with this system by orbiting the test gyroscope in the so-called Keplerian frequency around the non-rotating analogue black hole that can be constructed by making the rotation of the fluid/vortex negligible compared to its radial velocity.

**Keywords:** analogue gravity; black hole; gravity wave; spin precession

## 1. Introduction

Analogue black holes (BHs) obey similar equations of motion which are around a real BH that increase the possibility of demonstrating some of the most unusual properties of real BHs in the laboratory. The basic idea of such an analogue BHs (dumb holes) was originally proposed by Unruh [1] in 1981. The acoustic horizon of this type of analogue BH occurs when the fluid velocity exceeds the sound speed within the fluid. In 2002, Schützhold and Unruh [2] proposed another interesting model of analogue BH using the gravity waves of a flowing fluid in a shallow basin that can be used to simulate certain phenomena around BHs in the laboratory. In fluid dynamics, the gravity waves basically refer to the waves generated in a fluid medium or at the interface between the two media when the force of gravity or buoyancy tries to restore the equilibrium. For example, an interface between the atmosphere and the ocean gives rise to the wind waves. However, as the speed of the gravity waves and their high-wave-number dispersion can be adjusted easily by varying the height of the fluid and its surface tension, this new analogue BH model [2] has many advantages compared to the acoustic and dielectric analogue BH models. This new analogue BH model [2] can be used to simulate the various *astrophysical* phenomena in the laboratory to verify the prediction of Einstein's general relativity (GR).

We are now in an era to check the validity of GR in the strong gravity regime which is the most interesting and exciting regime. The spin precession of a test gyroscope due to the Lense–Thirring (LT) effect [3] as well as the geodetic/de-Sitter effect [4] is one of the predictions of GR, by which its validity in the strong gravity regime can be tested. It had already been verified in the vicinity of Earth, i.e., in the weak gravity regime with the Gravity Probe B space mission [5]. However, the direct measurement of the spin precession

effect in the strong gravity regime is extraordinarily difficult—far beyond the ability of current technology. Analogue models of gravity could be helpful in this regard. Creating an analogue (non-)rotating BH in the laboratory, one might be able to test the effect of spin precession in the strong gravity regime. Note that the geodetic/de-Sitter precession arises due to the non-zero curvature of spacetime and the LT precession arises due to the non-zero angular momentum of spacetime. Therefore, although the geodetic effect is observed in the *non-rotating* BH, the LT effect is absent there.

Specializing to the draining sink (DS) acoustic BH, the exact spin precession frequency due to the analogue LT effect has been predicted in the laboratory [6] which is commensurate with the prediction of Einstein's general relativity. Subsequently, the general expression for exact spin precession frequency in a stationary and axisymmetric spacetime was derived in [7] and applied to the $(2+1)$D DS acoustic BH [8] which is also commensurate with the Kerr BH. Although it was proposed [8] that the analogue spin precession frequency could be measured with the phonon spin (which acts as a test gyroscope in the DS BH set-up), the experimental set-up is quite complicated. To dispose of that complication, we propose a different experimental set-up in this paper which is easier to construct in the laboratory. Considering the gravity wave analogues of BHs [2], in this paper, we derive the exact spin precession frequency and propose how the measurement is feasible using a handy experimental set-up. Note that, the aim of most of research papers of analogue gravity is to measure the analogue Hawking radiation [1] or superradiance [9] in the laboratory with the help of analogue BHs. In contrast, our aim here is to measure the strong gravity effect on the spin precession in the laboratory with the help of rotating analogue BHs, as predicted [7,10] for the Kerr spacetime.

The paper is organized as follows. We recapitulate the model of gravity wave analogue of BHs proposed by Schützhold and Unruh in Section 2. For completeness, we revisited the general spin precession formalism in the $(2+1)$D analogue BH spacetime in Section 3. We derive the exact expression for spin precession frequency in the gravity wave analogue BHs in Section 4. The observational set-ups and prospects are discussed in Section 5. We conclude in Section 6.

## 2. Gravity Wave Analogue of (Non-)Rotating Black Hole

### 2.1. Recapitulation of Non-Rotating Black Hole

We begin with the simplest idea, i.e., the non-rotating analogue BH model with the gravity wave. We follow the same assumptions and the same model proposed earlier [2], i.e., a shallow liquid over a flat, horizontal bottom (see Figure 1). We assume that the forces on the liquid are such that they allow a purely horizontal stationary flow profile resulting in a constant height (that also means the horizontal surface) of the liquid. We also assume that the liquid is inviscid and incompressible in its flow. Thus, the density of the liquid remains constant ($\rho = $ const) and in terms of its local velocity $v$, the continuity equation assumes the simple form:

$$\nabla \cdot v = 0. \tag{1}$$

If we neglect the viscosity of the fluid, its dynamics are governed by the non-linear Euler equations (see Equation (2) of [2]). As we consider the exactly same model of the non-rotating analogue BH, we do not repeat the whole formulation discussed earlier [2] here. After some calculations (see Sections II–IV of [2] for the basic equations), the effective metric of the non-rotating analogue BH in the static Schwarzschild form as obtained earlier is given by

$$ds^2_{\text{nR}} = -\left(c_B^2 - w^2\right)d\tilde{t}^2 + \frac{c_B^2}{c_B^2 - w^2}dr^2 + r^2 d\varphi^2, \tag{2}$$

where $c_B = \sqrt{gh_B}$ is the speed of the gravity waves, $g$ is the acceleration due to gravity and $h_B$ is the height of the steady fluid (see Figure 1). The horizon occurs for $w = c_B = \sqrt{gh_B}$, i.e., when the velocity of the radially flowing fluid equals to the speed of the (long) gravity

waves. The trapping of the waves occurs inside this analogue horizon, when the radial velocity of the flowing fluid exceeds the speed of the gravity waves, i.e., $w > c_B$. The continuity equation implies [2,11]:

$$w(r) = \frac{C}{r}. \tag{3}$$

Note that the inward fluid flow with $w > 0$ corresponds to a BH, whereas the outward flow with $w < 0$ simulates a white hole. In this paper, we concentrate only on the BH.

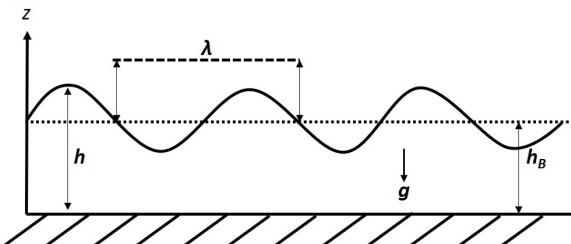

**Figure 1.** Schematic diagram of a gravity wave of the flowing fluid in a shallow basin, with the assumption of a long gravity wave ($h_B << \lambda$), where g is the acceleration due to gravity.

*2.2. Rotating Black Hole*

A stationary flow containing a component in the $\varphi$ direction, i.e., the vortex solution can be used to model a rotating analogue BH. Here, we consider a shallow liquid over a flat, horizontal bottom with a vortex. The gravity waves of the liquid with a vortex in the shallow basin can be used to simulate the phenomena around the rotating BHs in the laboratory. The centre of the vortex can be regarded as the centre of the rotating BH. Note that the centre, i.e., the singularity of the rotating analogue BH, is not like the ring singularity of the Kerr BH. However, using this analogue gravity model, one can measure the strong gravity effects in the laboratory which arise due to the rotation of the BH. In the case of the rotating analogue BH, the flow is characterized by the velocity potential [2]:

$$\vec{w} = \frac{C}{r}\,\hat{r} + \frac{L}{r}\,\hat{\varphi}\,, \tag{4}$$

where $(r, \varphi)$ are plane polar coordinates. The two parameters, $C$ and $L$, are constants, as well as analogous to the mass and angular momentum of a rotating BH, respectively. Now, using the flow of a barotropic, inviscid fluid with a vortex, one can write the rotating analogue BH metric in $(2+1)$D as [2]

$$ds^2 = -\left(c_B^2 - \frac{C^2 + L^2}{r^2}\right)dt^2 - 2\frac{C}{r}\,dt\,dr - 2L\,dt\,d\varphi + dr^2 + r^2 d\varphi^2. \tag{5}$$

The properties of the metric are clearer in a slightly different coordinate system defined through the transformations of $t$ and $\varphi$ coordinates [9], given by

$$dt \to dt + \frac{|C|r}{(r^2 c_B^2 - C^2)}dr \quad,\quad d\varphi \to d\varphi + \frac{|C|L}{r(r^2 c_B^2 - C^2)}dr. \tag{6}$$

In the new coordinates, Equation (5) reduces to, after rescaling the 'time' coordinate by $c_B$[1]:

$$ds^2 = -\left(1 - \frac{C^2 + L^2}{r^2}\right)dt^2 + \left(1 - \frac{C^2}{r^2}\right)^{-1}dr^2 - 2L\,d\varphi dt + r^2\,d\varphi^2 \tag{7}$$

(see Equation (6) of [9] or Equation (2) of [6]). Equation (7) effectively reduces to Equation (2) for $L = 0$ (and $c_B = 1$). The static limit $g_{tt} = 0$ denotes the region beyond which no particles can be in rest. This corresponds to the ergosurface, the radius of which is $r_e = \sqrt{C^2 + L^2}$. The velocity of the fluid is equal to the velocity of the gravity wave at $r = r_e$. The horizon

is defined by $r_h = C$ where the radial velocity of the fluid is equal to the velocity of the gravity wave. The region between $r_e$ and $r_h$ is known as the ergoregion, similar to the Kerr BH. If one defines a vertical coordinate $z$ as orthogonal to the bottom of the container, we obtain:

$$ds_3^2 = ds^2 + dz^2. \tag{8}$$

This is necessary to define the spin axis of the rotating analogue BH. The vertical axis is not important to study the analogue Hawking radiation or superradiance as the spin axis does not play any role in those cases. In stark contrast, the vertical axis plays a major role to study the analogue spin precession effect, as the axis of spin precession corresponds to the vertical $z$ axis. Note that although the metric Equations (2) and (7) look alike with the acoustic analogue BH metric (dumb holes) [6], the physical pictures are completely different.

### 3. Revisiting the Spin Precession Formulation in the Analogue BH Spacetime

In a rotating spacetime, a test spin/gyroscope can remain static without changing its location with respect to infinity outside the ergoregion. That is a static gyroscope whose four-velocity can be expressed as $u_{\text{static}}^\alpha = u_{\text{static}}^0(1, 0, 0, 0)$. On the other hand, if the gyroscope rotates (with an angular velocity $\Omega$) around the BH with respect to infinity, it is called a stationary gyroscope whose four-velocity is expressed as $u_{\text{stationary}}^\alpha = u_{\text{stationary}}^0(1, 0, 0, \Omega)$ [7]. The general spin precession frequency of a static gyroscope was derived earlier [12] and has been extended for a stationary gyroscope [7] for a general $(3+1)$D stationary and axisymmetric spacetime. The general spin precession formalism in a $(2+1)$D analogue BH spacetime was subsequently derived in [6,8] for static and stationary gyroscopes, respectively.

Now, let us consider that a test gyroscope/spin moves along a Killing trajectory in a stationary $(2+1)$D spacetime. The spin of such a test spin undergoes the Fermi–Walker transport [12] along the timelike Killing vector field $K$, whose four-velocity ($u$) can be written as [12]

$$u = (-K^2)^{-\frac{1}{2}} K. \tag{9}$$

In this special situation, the precession frequency of a test gyroscope coincides with the vorticity field associated with the Killing congruence, i.e., the gyroscope rotates relative to a corotating frame with an angular velocity. In a general stationary and axisymmetric spacetime, the timelike Killing vector is written as

$$K = \partial_t + \Omega \partial_\varphi, \tag{10}$$

where $\Omega$ is the angular velocity of the test gyroscope. The spin precession frequency of a test spin in $(2+1)$D spacetime can be derived as

$$\Omega_{(2+1)} = \frac{1}{2K^2} * (\tilde{K} \wedge d\tilde{K}), \tag{11}$$

following earlier work [12], where $\Omega_{(2+1)}$ is the spin precession rate of a test spin relative to a Copernican frame [12,13] of reference in a coordinate basis, $\tilde{K}$ is the dual one-form of $K$ and $*$ represents the Hodge star operator or Hodge dual. However, referring to the previous works [6–8] for detailed derivations, one can obtain the final expression for the spin precession frequency of a test gyroscope in $(2+1)$D spacetime as

$$\Omega_p = \frac{1}{2\sqrt{-g}(g_{tt} + 2\Omega\, g_{t\varphi} + \Omega^2\, g_{\varphi\varphi})} \left[ g_{tt}^2 \left(\frac{g_{t\varphi}}{g_{tt}}\right)_{,r} + g_{tt}^2\, \Omega \left(\frac{g_{\varphi\varphi}}{g_{tt}}\right)_{,r} + g_{t\varphi}^2\, \Omega^2 \left(\frac{g_{\varphi\varphi}}{g_{t\varphi}}\right)_{,r} \right]. \tag{12}$$

To preserve the timelike character of $K$ [7], i.e.,

$$K^2 = g_{\varphi\varphi}\Omega^2 + 2g_{t\varphi}\Omega + g_{tt} \quad < \quad 0, \tag{13}$$

$\Omega$ can only take those values which are in the following range:

$$\Omega_- < \Omega < \Omega_+ \tag{14}$$

where:

$$\Omega_\pm = \frac{-g_{t\varphi} \pm \sqrt{g_{t\varphi}^2 - g_{\varphi\varphi}g_{tt}}}{g_{\varphi\varphi}}. \tag{15}$$

Equation (14) covers a broad range of the orbital frequency for a test gyroscope including the zero angular momentum observer (ZAMO) and Kepler observer which play many important roles from the astrophysical point of view. Among all the possible orbital frequencies, the test gyroscope moves along the geodesic if and only if it orbits with the Kepler frequency. Otherwise, the test gyroscope will follow the other Killing trajectories and deviate from the geodesic. Hence, the gyroscope is accelerated. This is not unusual as a test gyroscope/spin does not follow a geodesic in general due to the spin–curvature coupling [14,15]. Additionally, there are also other reasons for which we do not restrict our study here to a test gyroscope that only moves along geodesics. In fact, the magnitude of the spin precession frequency is much larger for accelerating gyroscopes [7,10] compared to the non-accelerating gyroscopes that could be easily measured. The accelerated gyroscope can continue its stable precession very close to the horizon, whereas a non-accelerated gyroscope cannot continue its precession beyond the so-called innermost stable circular orbit (ISCO). In realistic astrophysical scenarios, there can be various sources of acceleration for a gyroscope which has been vividly described previously elsewhere [16].

### 3.1. Lense–Thirring Effect

Equation (13) reduces to the general expression for the LT precession frequency ($\Omega_p^{\text{LT}}$) in a $(2+1)$D rotating analogue BH spacetime [6] as

$$\Omega_p^{\text{LT}} = \frac{1}{2\sqrt{-g}}g_{tt}\left[\frac{g_{t\varphi}}{g_{tt}}\right]_{,\,r} \tag{16}$$

for $\Omega = 0$ which is only applicable for the static gyroscope outside the ergoregion. Equation (16) arises due to the presence of a non-vanishing $g_{0i}$ term, which signifies that the spacetime has an intrinsic rotation or a 'rotational sense' [13].

### 3.2. Spin Precession in the Vicinity of a Non-Rotating Analogue Black Hole

From our general expression Equation (13) of the spin precession, we identify that the arising spin precession in the spherically symmetric Schwarzschild spacetime is actually the so-called geodetic precession (see Section IV C of [7]) if the spin moves along the geodesic with the Kepler frequency. In this paper, the general expression for the spin precession in the spherically symmetric $(2+1)$D static spacetime can be written as

$$\Omega_{nR} = \frac{\Omega\, g_{tt}^2}{2\sqrt{-g}(g_{tt} + \Omega^2 g_{\varphi\varphi})}\left(\frac{g_{\varphi\varphi}}{g_{tt}}\right)_{,\,r} \tag{17}$$

where $\Omega$ is the angular velocity of the test gyroscope. If the test gyroscope moves along the geodesic, $\Omega$ can be regarded as the Kepler frequency ($\Omega_K$), given by

$$\Omega_K = \dot{\varphi}/\dot{t} = d\varphi/dt = \left[-\partial_r g_{t\varphi} \pm \sqrt{(\partial_r g_{t\varphi})^2 - \partial_r g_{tt}\,\partial_r g_{\varphi\varphi}}\right]/\partial_r g_{\varphi\varphi}. \tag{18}$$

### 4. Application to the Analogue BH Produced by Gravity Wave

Using Equations (7) and (13), one can obtain the spin precession frequency of a test gyroscope in the $(2+1)$D analogue gravity wave BH as

$$\Omega_p = |\Omega_p| = \frac{r_e^2(L - \Omega r^2) + \Omega r^2(r^2 - r_e^2) + L\Omega^2 r^4}{r^2[(r^2 - r_e^2) + 2L\Omega r^2 - \Omega^2 r^4]}. \tag{19}$$

In Equation (19), the test gyroscope can take any value of $\Omega$ between $\Omega_+$ and $\Omega_-$. One can introduce a parameter $q$ [7] to scan the range of allowed values of $\Omega$. Therefore, we can write from Equations (7) and (15):

$$\Omega = q\,\Omega_+ + (1-q)\,\Omega_- = \frac{1}{r^2}\left[(2q-1)\,\sqrt{r^2 - C^2} + L\right], \tag{20}$$

where $0 < q < 1$. Substituting $\Omega$ Equation (20) in Equation (19), one obtains:

$$\Omega_p = \frac{2L(1 - 2q + 2q^2)\sqrt{r^2 - C^2} - (1 - 2q)(r^2 - 2C^2)}{4q(1-q)r^2\sqrt{r^2 - C^2}}, \tag{21}$$

demonstrating that the spin precession frequency ($\Omega_p$) becomes arbitrarily large as it approaches to the horizon ($r \to C$) for all values of $q$ except $q = 1/2$ which corresponds to ZAMO. We discuss a few special cases in the next sections.

The evolution of the spin precession frequency ($\Omega_p$) of a test gyroscope can be seen from Figure 2 for three values of $q$: 0.2, 0.5 and 0.8. As we consider $C = L = 1$ (in the unit of length), the radius of the event horizon $r_h = C = 1$ and radius of the ergoregion $r_e = \sqrt{2}$. It can be seen in Figure 2 that $\Omega_p$ vanishes at a particular radius $r = r_0$ which is close to the horizon. The value of $r_0$ can be calculated using Equation (21) and by setting $\Omega_p|_{r=r_0} = 0$. This comes out as

$$r_0|_{(0<q<1/2)} = \frac{\sqrt{2Y}}{1 - 2q}\left[Y + L(1 - 2q + 2q^2)\right]^{\frac{1}{2}} \tag{22}$$

and:

$$r_0|_{(1/2<q<1)} = \frac{\sqrt{2Y}}{1 - 2q}\left[Y - L(1 - 2q + 2q^2)\right]^{\frac{1}{2}}, \tag{23}$$

where:

$$Y = \left[r_e^2\,(1 - 2q)^2 + 4q^2 L^2\,(1 - q)^2\right]^{\frac{1}{2}}. \tag{24}$$

Equation (22) is valid for $0 < q < 1/2$ and Equation (23) is valid for $1/2 < q < 1$. As $\Omega_p$ does not vanish in any radius for $q = 1/2$, $r_0$ does not arise for this specific case. In cases of $0 < q < 1/2$ and $1/2 < q < 1$, $\Omega_p$ increases with decreasing $r$, attains a peak and then decreases to zero at $r = r_0$. With further decrement in $r$ (i.e., for $r_h < r < r_0$), $\Omega_p$ increases and attains a large value very close to the horizon. A close observation reveals that the general features of dashed magenta and dot-dashed red curves are similar. For $q = 0^+$ and $q = 1^-$, the spin precession frequency vanishes at $r_0 \to \sqrt{2r_e\,(r_e + L)}$ and $r_0 \to \sqrt{2r_e\,(r_e - L)}$, respectively, which are greater than $r_h$. This means that $r_0$ occurs outside the horizon for all values of $q$ except $q = 1/2$.

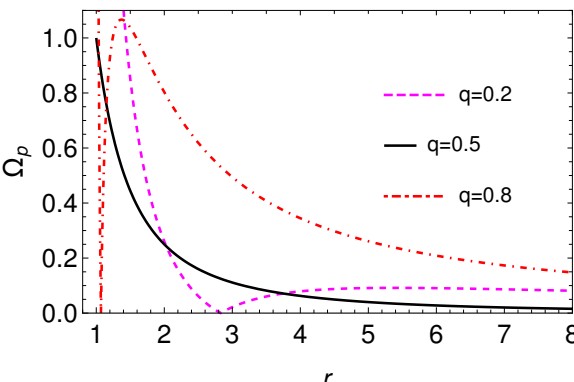

**Figure 2.** Rotating analogue BH: $\Omega_p$ (in the unit of length$^{-1}$) as a function of $r$ (in the unit of length) for $C = L = 1$ (i.e., $r_h = 1$ unit and $r_e = \sqrt{2}$ unit) and for three different values of $q$ which correspond to the different orbital frequencies of the test gyroscope. This figure shows that $\Omega_p$ first increases with the decrease in $r$ in the weak gravity regime, as usual. In the strong gravity regime, it attains a peak for the further decreasing of $r$, then decreases to zero at $r = r_0$. It again increases and becomes arbitrarily high very close to the horizon. This is true for all values of $q$ except $q = 0.5$ for which $\Omega_p$ monotonically increases for a decreasing $r$ and remains finite on the horizon.

The feature of all curves in Figure 2 is qualitatively similar to that of the curves in Figure 3 which are drawn for the Kerr BH with Kerr parameter $a_* = 0.5$ using Equation (51) of [7]. Figure 3 is drawn only for $\theta = \pi/2$ as the test gyroscope is considered to orbit in the equatorial plane. This is similar to the case considered here for the $(2+1)$D analogue BH spacetime. One can also look into Figure 3 of [7], where $\Omega_p$ is shown for the different values of the Kerr parameter with the different angles. The distance of the test gyroscope from the centre of the BH is plotted along the $x$ axis in Figure 3 in the unit of length $R_g{}^2$, whereas the precession frequency of the test gyroscope is plotted along the $y$ axis in the unit of $R_g^{-1}$.

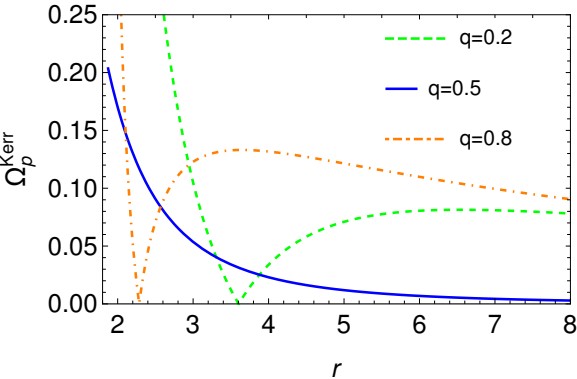

**Figure 3.** Kerr BH: Spin precession frequency $\Omega_p^{\text{Kerr}}$ (in the unit of $R_g^{-1}$) for the Kerr BH as a function of $r$ (in the unit of $R_g$) for $a_* = 0.5$, $\theta = \pi/2$, (with $r_h = 1.87R_g$ and $r_e = 2R_g$) and for different values of $q$ which correspond to the different orbital frequencies of the test gyroscope. This figure is drawn using Equation (51) of [7]. The qualitative features of all three curves resemble the three curves of Figure 2, which confirms that the spin precession effect in the analogue BH spacetime accurately mimics the spin precession effect near the vicinity of a Kerr BH.

*4.1. Lense–Thirring Precession: $\Omega = 0$*

For $\Omega = 0$ [17], Equation (19) reduces to Equation (9) of Ref. [6], given by

$$\Omega_p^{\text{LT}} = \frac{Lr_e^2}{r^2(r^2 - r_e^2)},\tag{25}$$

which is called as the Lense–Thirring (LT) precession frequency $(\Omega_p^{\text{LT}})$ [12]. It only arises due to the presence of $L$, i.e., only the rotation of the analogue BH is responsible for the

non-zero $\Omega_p^{\text{LT}}$. Therefore, one can see that the test spin precesses in the fluid with vortex due to the passing of gravity wave. Note that the LT precession frequency in Equation (25) is derived in the Copernican frame [12,13]. In this frame, the spin axis is primarily fixed at asymptotic infinity and the precession is measured relative to this frame.

### 4.2. Precession for the Zero Angular Momentum Observer: $q = 1/2$

If the angular momentum of a 'locally non-rotating observer' is zero, it is called a 'zero angular momentum observer' (ZAMO) which was first introduced by Bardeen [18,19]. Bardeen et al. [20], in fact, showed that the ZAMO frame is a powerful tool in the analysis of physical processes near the astrophysical objects. In our formalism, the ZAMO corresponds to $q = 1/2$ for which the angular velocity $\Omega$ reduces to:

$$\Omega_{\text{ZAMO}} = \frac{L}{r^2}. \tag{26}$$

It is interesting to see that the spin precession frequency remains finite, given by

$$\Omega_p|_{\Omega=\Omega_{\text{ZAMO}}} \to \frac{L}{C^2} \tag{27}$$

on the horizon in case of the ZAMO, unlike the other values of $q$, as seen from Figures 2 and 3. Note that the angular velocity ($\omega$) of the fluid is $\omega = L/r^2$ (see Equation (4)). Therefore, if a test gyroscope orbits with the same angular velocity of the fluid (i.e., $\omega = \Omega_{\text{ZAMO}}$), it could be able to cross the horizon with the finite precession frequency.

### 4.3. Geodetic/de-Sitter Precession: $\Omega = \Omega_K$

It is well known that the geodetic/de-Sitter precession arises due to the non-zero curvature of spacetime. It does not depend on the rotation of spacetime. Therefore, it can arise even in (non-rotating) Schwarzschild spacetime [17]. One can expect the analogue geodetic precession in the case of the non-rotating analogue BH spacetime produced by the gravity wave. However, the general spin precession expression which only arose due to the presence of analogue non-zero curvature [21] is derived in Equation (17). Now, using the non-rotating analogue BH metric (Equation (2) with $w = C/r$), one can deduce the spin precession frequency as

$$\Omega_{\text{nR}} = \Omega \, \frac{r^2 - 2C^2}{r^2 - C^2 - \Omega^2 r^4}, \tag{28}$$

where $\Omega$ is not necessarily a function of $r$; rather, it can take any finite value so that Equation (13) is satisfied [7]. For moving along the circular geodesic, $\Omega$ must be the so-called (analogue) Keplerian frequency, i.e., $\Omega_K = C/r^2$, which is derived using Equation (18). Substituting $\Omega = \Omega_K$ in Equation (28), one obtains:

$$\Omega_{\text{nR}}|_{\Omega=\Omega_K} = \frac{C}{r^2}. \tag{29}$$

As we discussed before, Equation (29) is derived in the Copernican frame. This means that this expression is computed with respect to the proper time $\tau$ which is related to the coordinate time $t$ via $d\tau = \sqrt{1 - \frac{2C^2}{r^2}} \, dt$. Finally, the precession frequency in the coordinate basis can be expressed as

$$\Omega_{\text{geo}} = \Omega_K - \frac{C}{r^2}\sqrt{1 - \frac{2C^2}{r^2}} = \frac{C}{r^2}\left[1 - \sqrt{1 - \frac{2C^2}{r^2}}\right], \tag{30}$$

which looks qualitatively similar to the expression for the original geodetic precession mentioned in Equation (14.18) from [17]. Equation (30) is considered for the geodetic

precession frequency ($\Omega_{\mathrm{geo}}$) in the gravity wave analogue BH, where the parameter $C$ plays the role of BH mass.

If one calculates the radial epicyclic frequency ($\Omega_r$) for the non-rotating analogue BH metric Equation (2), one obtains:

$$\Omega_r^2 = \frac{4C^4}{r^6}. \tag{31}$$

We know that the square of the radial epicyclic frequency ($\Omega_r^2$) vanishes at the innermost stable circular orbit (ISCO), and becomes negative for the smaller radius which leads the radial instabilities for those smaller radius orbits. Equation (31) reveals that the stable circular orbits exist everywhere in this analogue BH spacetime for any value of $r \geq r_h = C$. This leads us to conclude that the analogue geodetic precession continues for the orbits: $r \geq \sqrt{2}C$. No geodetic precession could be seen for $C < r < \sqrt{2}C$, as Equation (30) becomes imaginary. The similar situation does not arise in the case of the Schwarzschild BH, because the ISCO occurs at $r = 6R_g$, whereas the geodetic precession becomes imaginary for the orbits $r < 3R_g$. Therefore, the geodetic precession is meaningless for the orbits $2R_g < r < 6R_g$. As the above discussion also reveals, although the respective natures of the radial instabilities of the analogue BH and the real BH do not exactly resemble each other, the natures of the geodetic precession for both of the BHs match well.

## 5. Observational Prospects

The surface gravity wave is a region of increased and decreased depth that moves relative to the fluid. The large amplitude wave is that which produces a large change in depth compared to the mean depth as it goes by. If the change in depth is small compared to both the mean depth and wavelength, then it is called a small amplitude wave [22].

We have already mathematically shown the test spin/gyroscope precesses in the analogue BH spacetime due to the geodetic and LT effects. These effects could be measured using the gravity wave analogue BH produced in a shallow basin at the laboratory. Here, we consider two experimental shallow basin set-ups: (i) almost vortex-free and (ii) with vortex. Set-up (i) mimics the non-rotating analogue BH with which we can measure the analogue geodetic precession. On the other hand, we will be able to measure the LT precession as well as the total spin precession frequency with the arrangement of set-up (ii), which mimics a rotating analogue BH. It is important to note here that although we derived the various spin precession frequencies in the $(2+1)$D spacetime for simplicity, it can be converted to $(3+1)$D spacetime, as mentioned at the end of Section 2.

In this regard, one can define the spin axis along the $z$ axis which corresponds to the thread as shown in Figure 4. The modulus of the spin precession frequency is already derived in Section 4 and its direction is given along the $z$ axis (i.e., the thread). Thus, this whole arrangement can act as a test gyroscope to measure the spin precession frequency which arises due to the non-zero curvature and/or the non-zero rotation of the analogue BH. These effects were predicted long ago, but have not been measured to date in the strong gravity regime due to the lack of present technological expertise. However, using our proposed arrangement, it might be possible to measure the strong gravity spin precession effect in the laboratory.

In Figure 4, the fluid particles are displaced due to the passing of the gravity wave. We can detect/measure the spin precession effects by the motion of the fluid particles (see [23]). One way to see the motion of fluid particles is to introduce the neutrally buoyant solid particles into the fluid, as shown in Figure 4 and earlier [23]. The mean location of the solid particles will be stabilized by using the *slightly* buoyant particles [22], i.e., the specific gravity ($\rho_p$) of the particles should be slightly less than the specific gravity ($\rho_f$) of the fluid: $\rho_p < \rho_f$. One can mathematically express it as

$$\rho_p = \rho_f - \epsilon \tag{32}$$

where $\epsilon \to 0^+$. Fulfilling this condition is not only necessary for stabilizing the location of the particles, but also important for the spacetime point of view. Specifically, the original spin formalism [7] was developed considering a test spinning particle with the assumption that it does not distort the background BH spacetime. This assumption still holds in the case of the gravity wave analogue BH model considered in this paper, if Equation (32) is satisfied. However, the buoyant particles are also stabilized attaching a thread to them which extends to the bottom where a short-length miniature chain is attached. There are generally two types of fluid waves that can be observed [22]—one of which are deep fluid waves ($\lambda \lesssim h$). In this case, the particles move in (approximately) circular paths and the diameter of the circles decreases with depth [22]. The particles near the bottom hardly move at all [22]. The second one is the shallow fluid waves ($\lambda > h$) for which the horizontal amplitude of the particles motion is nearly same at all depths [22]. As we mentioned previously, we consider the long gravity waves or the shallow fluid waves (see Figure 1) in this paper.

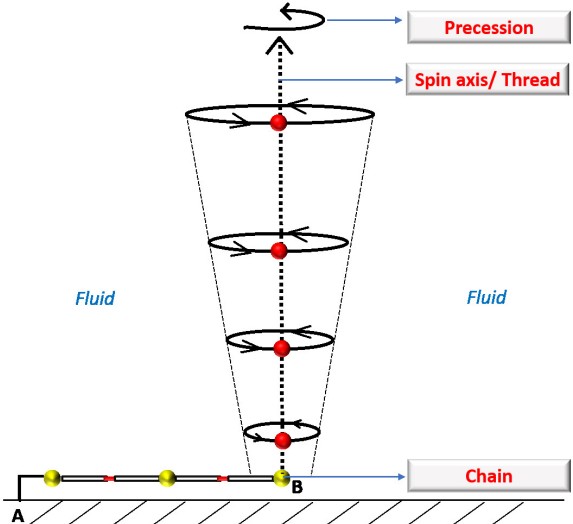

**Figure 4.** Schematic diagram of the arrangement in the laboratory to measure the spin precession frequency which arose from analogue BH spacetime. In this arrangement, the buoyant (red coloured) particles are attached with a thread and attached to the bottom of the fluid with a short-length miniature chain which is anchored at point A in the basement of the shallow basin. The chain is flexible, so that it can freely move in any direction to measure the spin precession. The whole system is used as a test gyroscope. The thread acts as the axis of the gyroscope.

Let us now consider our new test gyroscope (as shown in Figure 4) submersed in a fluid with a vortex, and rotating around the vortex with the orbital frequency, such that $0 < q < 1$. One should be able to regulate the orbital frequency of the test gyroscope by $q$. This helps one to measure the spin precession frequency for an accelerating test gyroscope as well. However, when the gravity waves pass by, the test gyroscope starts to process. The precession could be measured by observing the precession of the thread. The nature of the plots of Figure 2 could be visualized by shrinking the orbit of the test gyroscope. As the chain is flexible, the test gyroscope can orbit the vortex freely in any direction to measure the spin precession. Note that the uppermost portion of the thread, or, the uppermost buoyant particle attached to the thread could be ideal to measure the spin precession frequency, as it seems to be easier to visualize the effect of spin precession using that part of the test gyroscope.

If one wants to measure the spin precession frequency which only arose due to the LT effect, the test gyroscope has to be held/fixed ($\Omega = 0$) at a particular point. To achieve this, the point B of Figure 4 is anchored at the basement of the fluid container as in point A. This helps to freely process the thread (i.e., the spin axis) without disturbing the position of the whole test gyroscope.

In case of a rotating analogue BH, one could consider the vortex as $r \to 0$, i.e., the analogue of a BH singularity. In the case of the non-rotating analogue BH, one could not be able to define such a singularity. For example, the location of the acoustic horizon is defined at the narrowest point of a naval nozzle (see Figure 1.2 of [11]), as the fluid flow goes supersonic at that point. After coming out from that point, the fluid flow again becomes subsonic. Therefore, it is almost impossible to define such an analogous singularity in a non-rotating analogue BH. The similar situation arises in case of the non-rotating analogue BH produced by the gravity waves. In such a case, it is not easy to directly measure the analogue geodetic/de-Sitter precession by producing a completely non-rotating analogue BH. Moreover, the test gyroscope must orbit the analogue BH with the analogue Kepler frequency ($\Omega_K$) to measure the geodetic precession. Considering all these points, let us use the same set-up which we used for the rotating analogue BH (i.e., the fluid flow with vortex) with the condition $L << C$, so that the effect of $L$ can be neglected. In this approximate system (i.e., the almost vortex-free set-up that we have already mentioned in the beginning of this section), the test gyroscope (Figure 4) is able to orbit the vortex geometry with the orbital frequency $\Omega \to \Omega_K$, as we mentioned in Section 4.3. Thus, the measured spin precession frequency of the test gyroscope in this experimental set-up could be considered as the analogue geodetic/de-Sitter precession frequency.

## 6. Conclusions and Discussion

We derived the exact spin precession frequency of an accelerated test spin/gyroscope in analogue (non-)rotating BH spacetime. As a special case, one could measure the spin precession frequency for a non-accelerated test gyroscope (taking $\Omega = \Omega_K$) that moves along the so-called geodesic. Controlling the value of $\Omega$, one can measure the various spin precession frequencies which arise due to the LT effect, de-Sitter effect and also due to the non-zero angular velocity [21] of the test spin. Remarkably, we are able to construct a proper test gyroscope in a very simple way, by which it could be possible to measure these various spin precession frequencies in the laboratory by using a handy experimental set-up. We also showed that the spin precession frequency curves qualitatively follow the same trend, as it follows in the vicinity of the real Kerr BH. Therefore, measuring the spin precession effect in the laboratory by the gravity wave analogue BHs accurately reflects the strong gravity effect of spin precession in the Schwarzschild as well as Kerr BHs.

The advantage of the gravity wave analogue BH is that it is possible to easily simulate the BH in the laboratory. The main advantage is the possibility of independently tuning the velocity of the gravity wave propagation and measuring its amplitude directly with a very high accuracy by controlling only the height of the background fluid. The same is not so easy in case of the sound waves for the draining bathtub spacetime. To properly mimic the original spin precession effect of the strong gravity regime in our laboratory, we should be able to control the orbital frequency (using the parameter $q$) of the test gyroscope. In the case of the draining bathtub spacetime [8], it is extremely difficult to define and/or construct a proper test gyroscope [8] to measure the spin precession effect in the laboratory. In contrast, we are able to construct a simple test gyroscope by the buoyant particles and define a proper axis to measure the spin precession. Note that the fluid acceleration can also occur if the height $h$ changes [22]. This also instigates the test gyroscope to automatically accelerate. Although we do not consider such a case in this paper, our general spin precession formalism can be valid for that case with the help of $q$.

It would be useful to note here that our whole formulation is developed based on the inviscid fluid [2]. The non-zero viscosity of the fluid leads to the Lorentz violation [24], and in that case, the analogue BH metric may not be extractable in a closed form [6]. Although Schützhold and Unruh suggested some practical remedies (see Section VI of [2]) to overcome this problem, it is still there. In spite of the problem related to the viscosity, the first laboratory detection of the 'superradiance' phenomenon was possible by using the water [25]. However, no such experiments have been performed to date in the laboratory for the detection of the strong gravity effect of the spin precession.

Note that the spin precession effects could also be analysed using the other analogue models of BHs based on the boson gas and/or neural networks [26–29]. For instance, if one constructs an acoustic analogue BH (i.e., a draining bathtub acoustic spacetime) with a Bose–Einstein condensed (BEC) system, the strong gravity effect of the spin precession could be measured by using the BEC gyroscope (see Figure 1 of [30]).

**Author Contributions:** The problem was triggered during a joint discussion among the authors. C.C. then took the initiative to formulate it based on a further literature survey. B.M. verified the formulation part and discussed doubts with C.C. Observational prospects and experimental design are performed by C.C. The first version of the manuscript was written by C.C. and it was subsequently revised by C.C. and B.M. together. All authors have read and agreed to the published version of the manuscript.

**Funding:** C.C. was supported by a fund from the Department of Science and Technology (DST-SERB) with research Grant No. DSTO/PPH/BMP/1946 (EMR/2017/001226) during this research.

**Institutional Review Board Statement:** Not applicable.

**Informed Consent Statement:** Not applicable.

**Conflicts of Interest:** The authors declare no conflict of interest.

## Notes

[1]  Note that the dimension of $C$ and $L$ in Equation (7) transforms to the dimension of length due to the rescaling, whereas the dimension of $C$ and $L$ in Equations (3)–(5) is length$^2$/time. We follow the structure of Equation (7) in the rest of this paper, i.e., all quantities are expressed in $c_B = 1$ unit.

[2]  $R_g = GM/c^2$ which is called the gravitational radius, where $M$ is the mass of the BH, $G$ is the Newton's constant and $c$ is the speed of light in vacuum.

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
