# Peer review of "Spin Precession in the Gravity Wave Analogue Black Hole Spacetime"

_universe, doi:10.3390/universe8030193_

Round 1
Reviewer 1 Report
The manuscript Chandrachur Chakraborty , Banibrata Mukhopadhyay ‘Spin precession in the gravity wave analogue black hole spacetime’ deals with BH rotation in space with constant sizes.
I am not specialist in gravitational waves or in black holes rotation. However the picture 1 doesn not take into account the changes of BH mass. Moreover the equation (1), Euler equations should be reobtained in the case of gravity interaction. Also I do not found several papers citated literature.
I suppose the manuscript can be publish after correction carried out according to the author's opinion or as it is if reduction suppose that this is possible
Reviewer 2 Report
In this article, the authors propose a laboratory setup to measure the strong gravity effect on the spin precession of a gyroscope. To measure such strong gravity effect, they use the gravity wave analogue black holes (BHs) produced in a shallow liquid over a flat, horizontal bottom at the laboratory. When a vortex is present in the liquid, a rotating analogue BH is modeled. A test gyroscope is constructed by attaching a thread with the buoyant particles. The thread acts as the axis of the gyroscope. This gyroscope is used to measure the spin precession frequency which arises due to the non-zero curvature and/or the non-zero rotation of the analogue BH.
The paper is well written. The ideas and corresponding analyses are well conveyed. I recommend the publication of this article in the journal. I would however appreciate if authors can provide additional discussions based on the following comments:
- Authors consider ideal inviscid fluid while performing the calculations and drawing their conclusions. However, practically it’s difficult to find such ideal fluid. Therefore, for a realistic laboratory setup, how much deviation from the idealistic results can one expect?
- Is it possible to provide any qualitative or quantitative comparison on the spin precision in strong gravity for real and analogue BH?
Reviewer 3 Report
The authors analyse the spin precession effects in analogue models of Black Holes. They consider that using analogue models is an advantage because the experiments can be carried out directly in a laboratory. The model which they use is based on a fluid flowing with a speed larger than the speed of sound inside it. The speed of sound is taken as the speed of a gravitational waves or as the speed of light. Although the paper is interesting, I would like the authors to complement what they have found with some impressions and possibly calculations with a different analogue model. If the authors could mention how to express the precession problems based on the case where we consider an analogue model of Black Hole based on a gas of bosons and/or neural networks, such as those presented in EPJ Quantum Technology 6 (2019) 1; EPL 124 (2018) 5, 50002; arXiv:1804.06154 [hep-th]; Phys.Rev.D84:024039, 2011, among other references. If the authors could at least mention how to address the precession issues in these additional analogue models based on condensed matter systems, then the paper would be more complete and it will catch more the attention of a general audience. After the authors address my comments, I will happily revise the paper again.
Round 2
Reviewer 3 Report
The authors have addressed my comments. The paper can be published in the present form.